

# Ship borne rotating shadow band radiometer observations for the determination of multi spectral irradiance components and direct sun products for aerosol

Jonas Witthuhn[1], Hartwig Deneke[1], Andreas Macke[1], and Germar Bernhard[2]

[1]Leibniz Institute of Tropospheric Research, Leipzig, Germany
[2]Biospherical Instruments Inc., San Diego, CA

*Correspondence to:* jonas.witthuhn@tropos.de

**Abstract.** The 19 channel rotating shadow band radiometer GUVis-3511 built by Biospherical Instruments is introduced as an instrument which is able to provide automated ship borne measurements of the direct, diffuse and global spectral irradiance components without a requirement for stabilization. Several direct sun products, including spectral direct beam transmittance, aerosol optical depth, Ångström exponent, and precipitable water can be derived from these observations. The individual steps
of the data analysis are described, and the different sources of uncertainty are discussed. The total uncertainty of the observed direct beam transmittances is estimated to be 4.24% at 95% CI for ship borne operation. The calibration is identified as the dominating contribution to the total uncertainty. A comparison of direct beam transmittance with those obtained from a Cimel sun photometer at a land site and a manually operated Microtops II sun photometer on a ship is presented, yielding relative deviations of less than 3% and 4%, on land and on ship, respectively, for most channels and in agreement with our previous
uncertainty estimate. These numbers demonstrate that the instrument is well suited for ship borne operation, and the applied methods for motion correction work accurately. Based on spectral direct beam transmittance, aerosol optical depth at 510 nm can be retrieved with an uncertainty of 0.0032 for a 95% CI. Only minor deviations occur due to the different methods used for estimating Rayleigh scattering and gas absorption optical depths, as implemented by AERONET and in our processing. Relying on the cross-calibration of the 940 nm water vapor channel with the Cimel sun photometer, the column amount of
precipitable water has been estimated with an uncertainty of ± 0.034 cm. More research is needed to estimate the accuracy of the instrument for low sun (solar zenith angles larger than 70°) and during periods with strong swell.





# 1 Introduction

Aerosol and clouds are important components of the Earth's climate system. Detailed knowledge of their interactions as well as their radiative properties and effects is crucial to advance our understanding of climate change (Boucher et al., 2013). One specific aspect which requires further clarification is their interaction with solar radiation through scattering and absorption, and the resulting modulation of the short wave radiation budget.

Focusing on aerosol, the Aerosol Robotic Network (AERONET) provides a relatively dense observational network of aerosol optical depths (AOD) and further properties retrieved from Cimel sun photometers (Holben et al., 1998) over land. The Multi-filter Rotating Shadowband Radiometer (MFRSR) established by the U.S. Department of Energy's Atmospheric Radiation Measurement (ARM) Climate Research Facility is another widely used instrument to measure spectral irradiance components, aerosol and cloud optical properties (Harrison et al., 1994; Hodges and Michalsky, 2011).

Over ocean, however, our knowledge about aerosol properties and climatology is limited due to the low density of observations (Haywood et al., 1999). Compared to the techniques used over land, ship borne observations are also more challenging due to the continuously moving nature of the platform caused by waves.

To address this point, the Maritime Aerosol Network (MAN) has been established as a sub project of AERONET. It uses hand held Microtops II sun photometers (referred as Microtops in the following text), and thus relies on the skill of human observers to compensate for the ship movement (Smirnov et al., 2009). Using sun photometers on stabilized platforms is one alternative, but requires highly complex hardware, which so far is too expensive for wide spread use. The shadow band radiometer offers a promising alternative to the stabilization or manual tracking of sun photometers for ship-born operation, if a constantly moving shadow band is used (Reynolds et al., 2001). In addition, it provides direct information about radiative fluxes and thus aerosol and cloud radiative effects. This type of radiometer observes spectral irradiance with a high sampling frequency, while a shadow band sweeps across the upper hemisphere and causes a well defined shadow to fall on the sensor during its transit. From this time series, it is possible to identify the measurements when the sun is blocked, and to estimate the direct component of solar radiation even if the platform (eg. the ship) moves, as long as the departure from the horizontal orientation is known.

The simultaneous measurements with the shadow band radiometer of aerosol optical properties and radiative fluxes avoids inconsistencies in calibration which are unavoidable if multiple detectors are used. Aerosol size distributions can be obtained from the spectral dependence of the AOD (King et al., 1978). High frequency sampling combined with a narrow shadow band can offer additional information about the distribution of circum solar radiation, and can be exploited to retrieve cloud optical depth and effective radius (Min and Duan, 2005; Bartholomew et al., 2011).

Within the frame of the OCEANET project (Macke, 2009), a ship borne facility was developed for long term investigation of the transfer of energy, particles and chemical compounds between ocean and atmosphere. Since 2009, twelve cruises have been conducted with detailed atmospheric measurements on the German research vessel *Polarstern* during its meridional transfer cruises between the hemispheres, including aerosol observations as part of MAN. To improve and extend observational capabilities, a GUVis-3511 radiometer (referred as GUVis in the following text) was acquired in 2014 from Biospherical



Instruments Inc. (BSI), which is equipped with a constantly moving shadow band accessory termed BioSHADE (Morrow et al., 2010). The radiometer offers 18 spectral channel ranging from 305 nm to 1640 nm, and includes all AERONET and MFRSR channels as well as and some additional wavelength bands. This wide spectral range and the ability to measure on a ship makes this instrument and its data products unique and will enable to gain further insight into the properties and radiative effects of aerosol over the ocean.

The goals of this paper are threefold: First, we present the GUVis shadow band radiometer and the algorithms implemented at TROPOS for the data analysis. This includes the calculation of the spectral irradiance components including a motion correction for operation on ships, and the subsequent retrieval of spectral AODs, Ångström coefficients and atmospheric water vapor column from the direct irradiance measurements (direct-sun products). Secondly, an uncertainty analysis of these products is given based on theoretical considerations. Finally, a comparison is presented with a Cimel sun photometer over land and Microtops observations over sea, to confirm our accuracy estimates and the reliability of the products.

A detailed instrumental description is provided in Sect. 2. We present an overview of the current data processing and analysis for the GUVis observations based on the direct irradiance measurements in Sect. 3. A detailed analysis of the observational uncertainties is given in Sect. 4. The comparison of these observations with land and ship borne observations is given also in this section. The paper ends with a summary and an outlook in Sect. 5.

## 2 Instrumentation

The GUVis radiometer is a multi channel filter instrument (Seckmeyer et al., 2010) with 18 spectral channels, ranging from 305 to 1640 nm with a bandwidth of approximately 10 nm. Each channel consists of interference and blocking filters (e.g., UG-11 and BG-25 bandpass filters from Schott) that are coupled to a "microradiometer" (Morrow et al., 2010). Each microradiometer includes a photo detector, pre amplifier with 3-stage gain, 24 bit analogue-to-digital converter, microprocessor, and an addressable digital port. Data streams from all microradiometers are combined with measurements from ancillary sensors (e.g., temperature) and transmitted via a USB port to a PC. The design does not require to multiplex analogue signals from multiple photo detectors, resulting in less electronic leakage and better reliability than traditional approaches. The instrument's internal temperature is stabilized to $40 \pm 0.5\,°C$ using a proportional-integral-derivative controller. Silicon photo diodes are used for channels with wavelengths up to 1020 nm, while channels above this wavelength use indium gallium arsenide detectors. Channels were selected from a list of standard wavelengths equipped with hard-coated ion-assisted deposition interference filters, which are known for excellent long-term stability. For the TROPOS instrument, three custom wavelengths were chosen to optimize the information content for atmospheric retrievals, and had to be realized using less durable soft-coated interference filters for cost reasons. Specifically, this applies to the channels at 750 nm as absorption-free reference for the 765 nm Oxygen-A band channel, the 940 nm channel to measure the atmospheric water vapor column (Halthore et al., 1997), and the 1550 nm channel for cloud micro physics retrievals (Brückner et al., 2014). Data analysis suggests that the transmission of these soft coated filters has changed significantly during the deployment of the instrument (Sect. 2).



The filter microradiometer assemblies point at the center of an irradiance collector, which features a composite diffuser made of layers of generic and porous polytetrafluoroethylene (PTFE) sheets (Hooker et al., 2012). This design leads to relatively small cosine errors also in the infrared, where the scattering properties of traditional PTFE diffusers are typically degraded. The instrument is also equipped with two orthogonally-mounted accelerometers for determining the instrument's inclination (pitch and roll). The two sensors are not designed for use in a dynamically moving environment, such as on ships, and measurement errors will occur when the instrument's orientation is changing rapidly.

The radiometer is equipped with a computer controlled shadow band accessory, called BioSHADE (Morrow et al., 2010). The band is made of anodized black aluminium, is $2.5\,\mathrm{cm}$ wide and has a diameter of $26.7\,\mathrm{cm}$. The GUVis typically samples at $15\,\mathrm{Hz}$ when the band is moving. The band rotates at a constant speed such that at least 5 data points are sampled during the time when all parts of the diffuser are shaded by the band. For measuring global irradiance, the band is stowed below the horizon of the instrument's diffuser. The instrument is also equipped with a GPS receiver, called BioGPS, which determines latitude, longitude and time once per second, and adds this information to the data stream. The GUVis is controlled by a data acquisition software running on a Windows laptop, which records the raw sensor signals, converts these to spectral irradiances for all channels by applying calibration coefficients stored internally in the instrument, and records the irradiance plus additional status information in ASCII data files.

The instrumental set up is shown in Fig. 1 where the GUVis is mounted on the research vessel *Polarstern* together with a total sky imager, which is used to identify sky conditions, and as supplementary information for interpreting the irradiance measurements.

**Uncertainty of the calibration**

The instrument was calibrated by the manufacturer at the time it was built. It was calibrated a second time after two years, to verify the stability of the instrument. For these calibrations, NIST-traceable 1000 Watt FEL standard lamps have been used. Table 1 shows the deviation of the calibration constants between both calibrations. Most channels show a deviation of less than $2\%$, which is within the expected range for the temporal drift of such an instrument.

The channels at 305, 340, and to a lesser extend at $380\,\mathrm{nm}$, show large drifts. These have been attributed to a change in the transmission of a special insert below the instrument's main Teflon diffuser, which is necessary to get an adequate cosine response at wavelengths larger than about $800\,\mathrm{nm}$. BSI has addressed this problem by replacing this insert with a new material in our GUVis instrument. Hence, the stability of these channels should be significantly improved in the future, which nevertheless needs to be verified by future calibrations.

The channels at 750, 940 and $1550\,\mathrm{nm}$ also show large deviations. They correspond to the custom channels chosen by TROPOS as mentioned in Sect. 2. These channels use soft-coated interference filters for cost reasons, which have a known lower temporal stability than hard-coated ones, as is confirmed by these findings. To improve stability, the filters could be replaced by hard-coated filters, which are however expensive in particular if they are ordered in small numbers. At this stage, no replacement of filters is planned for our instruments, and a low calibration uncertainty can only be achieved by frequent





calibrations. For the 940 nm channel, this can be realized by cross-calibration with AERONET observations in the field, as outlined in Sect. 3.3.

## 3 Method

Raw data are calibrated with calibration coefficients stored in the instrument's internal memory. The calibration has been performed by the manufacturer and includes an absolute calibration, a characterization of the sensor's deviation from the desired cosine response, and the determination of the spectral transmission of filters in the laboratory. These calibration data were shipped with the instrument and are used for our calculations and corrections.

For retrieving the direct irradiance and AOD, we have implemented several subsequent algorithms for data processing. These programs provide the separation of the irradiance components as well as the calculation of the spectral AOD. To achieve this, we use the proportionality of the observed direct normal spectral irradiance (DNI, $I(\lambda)$) to the spectral direct beam transmittance $T(\lambda)$ expressed by the Beer-Lambert Law (Beer, 1852), which is the fundamental relation exploited also by sun photometer observations:

$$I(\lambda) = \frac{I_0(\lambda)}{R_E^2} \exp\left(-m_a \tau(\lambda)\right) \tag{1}$$

$$I(\lambda) = \frac{I_0(\lambda)}{R_E^2} T(\lambda) \tag{2}$$

For the top-of-atmosphere (TOA) solar irradiance $I_0$ at one astronomical unit, the NewGuey2003 spectrum (Gueymard, 2004) is applied, which is convolved with the spectral response function of the GUVis channels obtained from the manufacturer's instrumental characterization. The $I_0$ is scaled by the inverse square of the actual sun earth distance normalized to one astronomical unit ($R_E$), which is calculated using equations given in WMO (2010). We assume the air mass factor $m_a$ to be equal to the inverse cosine of the zenith angle $\mu_0^{-1}$ here. The deviation from more complex expressions will be small as we are currently not using data with the sun below a zenith angle of 70° (see Sect. 3.2).

During processing the data is corrected for ship motion and cosine error of the instrument's irradiance collector. For AOD calculations, the OD for several atmospheric gases and Rayleigh scattering are taken into account. Also the time series is screened to exclude cloud contaminated data. The implemented methods are based on the description given previously by Morrow et al. (2010), Bannehr and Schwiesow (1993), Smirnov et al. (2000) and Alexandrov et al. (2002; 2007; 2008). In the following, the steps of our data analysis are described. An outline of the processing is given by the flowchart shown in Fig. 2.

### 3.1 Motion and cosine error correction

Motion and cosine error corrections are applied simultaneously in the first step, because of their interdependency.

The motion correction compensates for the levelling errors of the instrument due to the ship movement, and estimates the deviation from a horizontally aligned irradiance observation. This is crucial because the spectral irradiance is defined either





relative to a horizontal reference plane or a plane normal to the incidence angle of the sun. Due to the ship motion, the alignment of the instrument is changing continuously. This is compensated based on the method described by Bannehr and Schwiesow (1993), which was originally developed for pyranometer measurements on an air plane, but can be equally used for ship measurements. This correction factor ($C1$) is calculated from the ratio of the cosines of the true solar zenith angle($\Theta$)

and the apparent zenith angle ($\Theta_A$), which is calculated from the sun position and the ship's role, pitch and heading angles. The method from Bannehr and Schwiesow (1993) only corrects the direct irradiance component for the effects of motion, and is thus only applicable when the sun is visible. Due to anisotropy in the diffuse radiation field, e.g. due to Rayleigh scattering, also the diffuse component of irradiance changes with the tilt of the sensor. By adapting the method of Boers et al. (1998) and using radiative transfer calculations, carried out with the libradtran package using the DISORT solver (Mayer and Kylling,

2005), correction factors ($C2$ and $C3$) are calculated, which take diffuse radiation into account. This factors are defined by Boers et al. (1998) as:

$$C1(\Theta, \Theta_A) = \frac{\cos(\Theta)}{\cos(\Theta_A)} \tag{3}$$

$$C2(\Theta, \Theta_A, \lambda) = \frac{\cos(\Theta) + B(\lambda)}{\cos(\Theta_A) + B(\lambda)} \tag{4}$$

$$C3(\Theta, \Theta_A, \lambda) = \frac{\cos(\Theta) + B(\lambda) \cdot J(\Theta, \lambda)}{\cos(\Theta_A) + B(\lambda) \cdot J(\Theta_A, \lambda)} \tag{5}$$

where $B = \pi I_{dif}(\lambda)/I(\lambda)$ is the ratio of the diffuse ($I_{dif}(\lambda)$) to direct normal irradiance at the surface ($I(\lambda)$) with $\Theta = 0°$. $J(\Theta) = I_{dif}(\Theta, \lambda)/I_{dif}(\Theta = 0°, \lambda)$ is the diffuse irradiance retrieved by radiative transfer calculations assuming a clear sky with only molecular scattering (eg. Rayleigh scattering) at the solar zenith angle normalized to the diffuse irradiance at $\Theta = 0°$. The deviation of the three correction factors are shown in Fig. 3. For lower wavelengths $B$ is large and the diffuse irradiance becomes more dominant, therefore $C2$ of the 305 nm channel hardly follows $C1$. Because of the stronger Rayleigh scattering,

the diffuse irradiance at shorter wavelengths drops faster than the direct irradiance at lower sun elevation. Due to this effect, the deviation between $C1$ and $C3$ for channels with wavelengths around 300 nm have the largest values for solar zenith angles between 60° and 70° $\Theta$. The deviation is small and becomes less important for longer wavelengths due to the fact that Rayleigh scattering is almost negligible for wavelengths greater than 800 nm. Overall, except for short wavelengths around 300 nm, the deviation of the correction factors increases with decreasing sun elevation.

Assuming Rayleigh scattering to calculate the motion correction factors ($C3$) is considered to be the most realistic correction and is used in the presented algorithm, taken from pre-calculated lookup tables varying $\Theta$ and $\Theta_A$. Effects from aerosol are neglected in the radiative transfer calculations and the uncertainty resulting from this omission on the motion correction factor is investigated in the following subsection.

For measurements on *Polarstern*, data from the ship's marine inertial navigation system are used for motion correction. This

system provides precise measurements of the roll, pitch and heading angles of the ship at high temporal resolution. Because the instrument is not perfectly aligned relative to the ship's navigation system, we also apply a correction to account for this misalignment. This is done using the method of Bannehr and Schwiesow (1993), choosing data from clear days when the ship





moves while the sun is either in the front, back or the sides of the ship. In these cases, the tilt correction is dependent on either the roll or the pitch angles alone. For land operation, the instrument's position is static, but this correction is also applied using the instrument's internal accelerometer measurements to correct for slight misalignments of the set up. The internal measurements of pitch and roll angle have been calibrated using a precision level, and offsets relative to the diffuser are stored

in the instrument and corrected by the firmware.

When observing an inclined collimated beam from a horizontal plane with an ideal detector, the measured signal changes with the cosine of the incident zenith angle. The cosine error correction removes the deviation of the instrument's response for an inclined collimated incident beam of radiation from the ideal cosine response. The cosine error of the instrument is taken from a lookup table provided by the instrument manufacturer using $\Theta_A$ according to the ship motion. This lookup table has

been measured by the manufacturer individually for all spectral channels as part of the instrument calibration.

**Uncertainty of the motion correction**

The motion correction factor $C3$ is calculated by adapting the method of Boers et al. (1998) and using radiative transfer calculations which take Rayleigh scattering but no aerosol into account (AOD = 0). Calculations with aerosol need the knowledge of aerosol optical properties (e.g. size distribution, single scattering albedo, asymmetry parameter, optical depth) which we only

can guess at this stage of processing. To keep the processing fast, aerosol is neglected completely for the motion correction. To estimate the uncertainty of the motion correction we calculate correction factors using radiative transfer calculations taking aerosol with properties according to Shettle (1990) into account. The default properties are a rural type aerosol in the boundary layer, background aerosol above 2 km, spring-summer conditions and a visibility of 50 km. For our calculations the AOD is modified in the range of 0.05 to 0.45 comparing those correction factors to $C3$ without aerosol. Figure 4 demonstrates the error

of neglecting the aerosol while calculating the motion correction factor for the 305 nm and the 510 nm channel for $\Theta_A = \Theta$ - 6° (e.g. high swell). For higher apparent zenith angles (e.g. lower swell) the error will be reduced and turn negative when $\Theta_A > \Theta$.

From this calculations we estimate a motion correction uncertainty, forcing the AOD to be 0.45, which is a high AOD and rarely observed over ocean. Also the sky is assumed to be cloud free. The uncertainty is taken from pre calculated lookup tables depending on $\Theta_A$ and $\Theta$.

Applying a correction for aerosol and cloudy conditions requires additional information on the ratio of the direct to diffuse irradiance, and the anisotropy of the radiation field, which will be the subject of future investigations.

## 3.2   Separation of irradiance components

To calculate the irradiance components, the data of each shadow band sweep are analyzed separately. The irradiance is measured with a frequency of 15 Hz during the sweeps. With this temporal resolution, even short term irradiance fluctuations can

be resolved. The global irradiance is observed at the start and end of a shadow band sweep, when the shadow band is outside the field of view of the sensor. The minimum irradiance determined during the sweep correspond to the time when the diffuser is completely shaded by the shadow band, if the sun is visible. If no clear minimum is identified, the direct irradiance is very small or negligible, and only the global irradiance is determined by the algorithm.



The difference of the global irradiance and the minimum irradiance measured during the sweep represents the direct component of irradiance, together with an additional diffuse part blocked by the shadow band. Figure 5 shows an idealized time series for one sweep (red). The blue line represents the blocked diffuse irradiance during the sweep. This occurs because the shadow band blocks a significant part of the sky in addition to the sun. To estimate the amount of blocked diffuse irradiance,

30 data points before and after the transit of the shadow across the diffuser are used to extrapolate the diffuse irradiance to the point where the minimum is detected. Values from both extrapolations are averaged. With this information, we can calculate the direct irradiance as the difference of the global irradiance and the minimum irradiance, by subtracting the blocked diffuse part (Morrow et al., 2010). With lower sun and increased AOD load, the sweep minimum becomes less pronounced and it is more challenging to identify the shadow of the band on the sensor. The accuracy of extrapolations for these situations has to

be investigated in further work. At this stage we do not use observations with the sun lower than $70°$ zenith angle.

**Uncertainties caused by high-frequency fluctuations**

Noise in the electrical amplifiers of the radiometer directly affect the accuracy of the radiation measurements. We have attempted here to estimate the amplitude in each channel, using observations obtained during an absolute calibration in the laboratory. The amplitude is assumed to be constant for different levels of incident radiation. High-frequency fluctuations in

the direct beam transmittance during observations will introduce a similar uncertainty during our processing. Both effects are combined in the following uncertainty analysis.

The uncertainty due to noise is strongly reduced by averaging, which is in fact done several times by our method for separating the different irradiance components. The global irradiance is measured and averaged for 20 seconds between two sweeps, resulting in negligible uncertainty. The direct irradiance is however estimated using a smaller number of measurement

values. First, a mean irradiance is calculated while the diffuser is completely shaded from direct sun from at least five samples for clear sky, low AOD and high sun conditions and more than 10 samples for lower sun, which again reduces the influence of noise. Secondly, the shading of diffuse irradiance is estimated from the sweep data by linear extrapolation using 30 observations before and after the transit of the shadow across the diffuser. Uncertainties for the fit parameters are also calculated, which allow us to determine the uncertainty of the extrapolated values, and are attributed here to the influence of noise. Please note that

deviations from the underlying assumption of the linear model could also arise for other reasons, such as variations of the forward scattering peak, e.g. expected for large particles such as dust or ice crystals.

The uncertainties calculated for the DNI in this way during the Melpitz-Column experiment (see Sect. 4 for a brief description) do not exceed $0.6\%$ for a $95\%$ CI. Since the diffuse irradiance is calculated as difference of the direct and global irradiance and the uncertainty of the global irradiance due to measurement noise is negligible, its uncertainty is set to be equal to that for

the direct irradiance.

### 3.3  Calculation of OD

From the observed DNI, the OD $\tau$ of the atmosphere can be calculated from Eq. (1).





The cloud free atmospheric OD is influenced by Rayleigh scattering, aerosol and trace gas absorption. So far, we take absorption by ozone and $NO_2$ into account for all channels, and consider $H_2O$, $CO_2$ and $CH_4$ for channels matching those of the AERONET Cimel sun photometer (940, 1020 and 1640 nm). The AOD ($\tau_A$) can be determined by subtracting the Rayleigh ($\tau_R$) and trace gas absorption OD ($\tau_G$) from the total $\tau$ obtained from the measurements.

$$\tau_A(\lambda) = \tau(\lambda) - \tau_G(\lambda) - \tau_R(\lambda) \tag{6}$$

To calculate $\tau_R$ we have selected the method from Bodhaine et al. (1999), which takes pressure ($P$), $CO_2$ concentration ($CO_2$) and the gravitational constant depending on latitude ($lat$) and altitude ($alt$) into account. A current $CO_2$ global mean concentration of 400 ppm is assumed and local pressure observations are used.

Given the columnar number concentrations $n$ [$m^{-2}$] of $O_3$ and $NO_2$, the OD of these trace gases are calculated as:

$$\tau_{(O_3)} = \sigma_{A(O_3)} n \tag{7}$$

$$\tau_{(NO_2)} = \sigma_{A(NO_2)} n \tag{8}$$

$\sigma_A$ denotes the absorption cross section [$m^2$] of the gases and are taken from Schneider et al. (1987) for $NO_2$ and (Serdyuchenko et al., 2014) for $O_3$. Daily values of the columnar number concentration are obtained from the Aura - Ozone Monitoring Instrument (AURA-OMI) satellite data (McPeters et al., 2008; Bucsela et al., 2013).

For obtaining $CH_4$ and $CO_2$ absorption OD, estimates are obtained similarly to the sun photometer processing by AERONET. The absorption of $CO_2$ influences observations in both the 1550 nm and the 1640 nm channel, while the latter is also affected by $CH_4$ absorption. Based on computations using the standard US 1976 atmospheric model for the 1640 nm channel, the $CH_4$-OD was set to 0.0036 and the $CO_2$-OD to 0.0089 at a standard atmospheric pressure $P_0$ of 1013.25 hPa for the 1640 nm channel. Both ODs are then scaled with the actual air pressure $P$ by $\frac{P}{P_0}$.

The 940 nm channel is used to retrieve the precipitable water using a logarithmic transformation of the measured direct beam transmittance (Smirnov et al., 2004), where coefficients a and b in the following equation are instrument specific constants, and are usually determined by the filter response of the instrument (Pérez-Ramírez et al., 2014). Instead, we have chosen to obtain these coefficients from a fit of shadow band radiometer data to the precipitable water (w) obtained from the Cimel instrument by cross-calibration.

$$\mathrm{T}_{940,C} = \mathrm{T}_{940,w}\,\mathrm{T}_{940,A} = \exp\left(-a\left(\mathrm{w}\,m_w\right)^b\right)\exp\left(-\frac{\tau_A}{\mu_0}\right) \tag{9}$$

$$\ln\left(\ln\left(\frac{\mathrm{T}_{940,A}}{\mathrm{T}_{940,C}}\right)\right) = X = \ln(a) + b\ln(\mathrm{w}\,m_w) \tag{10}$$

$\mathrm{T}_{940,C}$ denotes the corrected transmittance for which transmittance from Rayleigh scattering and trace gases are already removed. Therefore $\mathrm{T}_{940,C}$ can be expressed as a product of water transmittance ($\mathrm{T}_{940,w}$) and aerosol transmittance ($\mathrm{T}_{940,A}$). The relative air mass factor for water vapor $m_w$ is calculated using the method of Kasten (1965). Using the GUVis 940 nm

channel and the Ångström exponent calculated from the 440 nm and 870 nm channels to estimate $\mathrm{T}_{940,A}$, we have determined values of $a = 0.6131$ and $b = 0.6712$ to best match the Cimel w.





With this approach, we avoid the use of spectroscopic data together with the filter response to establish the link between precipitable water and spectral direct beam transmittance. The advantage is that this ensures the consistency with the AERONET observations, and allows us to compensate for changes in transmittance of the 940 nm filter. The disadvantage is the reliance on AERONET observations.

5 The precipitable water is related linearly to the water vapor ODs $\tau_{\mathrm{w}}$ at 1640 nm and 1020 nm to account for water absorption in these channels (Schmid et al., 1996; Michalsky et al., 1995).

$$\tau_{\mathrm{w}}(1640\,nm) = 0.0014 \cdot \mathrm{w} - 0.0003 \tag{11}$$

$$\tau_{\mathrm{w}}(1020\,nm) = 0.0023 \cdot \mathrm{w} - 0.0002 \tag{12}$$

Due to the reliance on the Cimel channels, we cannot estimate the water vapor OD for the 1550 nm channel with this approach, 10 and are planning to carry out spectroscopic calculations for this channel in the future.

Comparing results obtained with our method from the GUVis instrument to the AERONET derived precipitable water, a close agreement with a standard deviation of only 0.029 cm (see Fig. 6) is found. Therefore, we conclude that this method is reliable as long as the calibration of the 940 nm channel remains stable, or collocated AERONET measurements can be used for cross-calibration.

### 15 Uncertainties of the Rayleigh OD

Since the calculation of the Rayleigh OD is direct proportional to the pressure, the uncertainty is given as:

$$\Delta \tau_R = \tau_R \frac{\Delta \mathrm{P}}{\mathrm{P}} \tag{13}$$

$\Delta\mathrm{P}$ is defined by the manufacturer of the weather station Lufft as $\pm 5\,\mathrm{hPa} \approx 0.5\,\%$.

This method assumes a current $CO_2$ concentration of about 400 ppm, which can vary over time. However, the deviation of 20 $\tau_R$ for variate $CO_2$ concentration of up to 40 ppm difference is only about 0.003 % and therefore negligible.

### Uncertainties of the O$_3$- and NO$_2$-OD

Because of the spectral characteristics of absorption, trace gases introduce a wavelength dependent uncertainty in the calculation of AOD. This uncertainty is mainly determined by the uncertainty of the trace gas column densities, which are obtained here from satellite retrievals by the AURA OMI instrument. The uncertainty of the column density of $O_3$ is set to 3%, and for 25 $NO_2$ column density to 20% as given in the OMI ATB (Bhartia, 2002; Chance, 2002). These uncertainties is directly translated into an uncertainty of the gas absorption optical depth, with different importance for different channels due to the wavelength dependence of both aerosol properties and gas absorption.

### Uncertainties of remaining gas absorption

The precipitable water is calculated using the 940 nm channel measurements in Eq. (10). From the linear regression of water 30 derived from both the Cimel and GUVis shown in Fig. 6 we found the standard deviation $\sigma_{\mathrm{w}}$ of 0.029 cm. From the Cimel




sample data uncertainty estimate (Holben et al., 1998) we estimate the standard deviation as $\sigma_C = 0.017\,\mathrm{cm}$. From this values we estimate the uncertainty of precipitable water $\Delta\mathrm{w}$ observed with the GUVis as:

$$\Delta\mathrm{w} = \sqrt{\sigma_\mathrm{w}^2 + \sigma_\mathrm{C}^2} = 0.034\,cm \qquad (14)$$

With this we calculate the uncertainty $\tau_\mathrm{w}$ for the 1020 and 1640 nm channel from Eq. (11) and Eq. (12) as $\Delta\tau_\mathrm{w}(1020\,\mathrm{nm}) = 7.82 \cdot 10^{-5}$ and $\Delta\tau_\mathrm{w}(1640\,\mathrm{nm}) = 4.76 \cdot 10^{-5}$.

The OD of $CO_2$ and $CH_4$ are scaled to the ambient pressure and applied only to the 1640 nm channel. Therefore, the uncertainties $\Delta\tau_{CO_2}$ and $\Delta\tau_{CH4}$ can be calculated from the uncertainty of the pressure measurements, assumed to have a value of $\Delta P = 5\,\mathrm{hPa}$. This leads to errors of $\Delta\tau_{CO_2}(1640\,\mathrm{nm}) = 4.361 \cdot 10^{-5}$ and $\Delta\tau_{CH4}(1640\,\mathrm{nm}) = 1.764 \cdot 10^{-5}$ for $CO_2$ and $CH_4$, respectively.

## 3.4 Cloud mask and quality control

To exclude cloud contaminated data from the calculation of aerosol properties, we have implemented a cloud mask algorithm as last processing step. Since the temporal resolution of the GUVis instrument is close to that of the Cimel sun photometer, we carry out the same procedure as described by Smirnov et al. (2000). The time series passes through three processing steps. In the first step, negative AOD values are removed, which may be caused by uncertainties in the correction for Rayleigh or gas absorption during low AOD conditions. The next step identifies triplets of data points with a variability greater than 0.02 in AOD as cloudy, which assumes that the AOD in the total atmospheric column is less variable than this threshold. The last step is a smoothness test, where the time series is compared against a smoothness criterion, and outliers are iteratively removed until the criterion is fulfilled (Smirnov et al., 2000). After this procedure, cloud contaminated and erroneous data points should be excluded from the subsequent calculation of AOD. Sample validations have been performed by comparing the clear sky identification with sky images from the total sky camera shown in Fig. 1. For the validations we chose samples from Melpitz-Column observations (see Sect. 4 for a brief description) at 16.06.2015, where fast changing and broken cloud situations are observed. In all cases no clouds are closer than 15° solid angle around the sun and for single samples the cloud cover reaches up to 0.5. Using the camera in addition to the GUVis we will improve the clear sky identification in further work.

## 4 Evaluation

The combined uncertainty of GUVis observations with respect to the observation of spectral horizontal irradiance and the estimation of the AOD is investigated in this section. The Melpitz-Column experiment took place in May to July 2015 in a rural area at the TROPOS measurement site Melpitz near Leipzig in Germany. During this time a complex set of aerosol and boundary layer measurements are installed to investigate the aerosol distribution in the whole tropospheric column. To verify the reliability of the GUVis shadow band radiometer, it has been deployed during the Melpitz-Column field experiment on land together with a Cimel sun photometer participating in the AERONET network, which allows a direct comparison of the observations and products.



As the main strength of the GUVis is its ability to be used on ships, measurements during the cruise PS83 with the RV *Polarstern* are also analyzed here and compared to MAN observations with a Microtops sun photometer.

## 4.1 Uncertainty estimation

We combine all the uncertainties mentioned so far to obtain the total AOD uncertainty ($\Delta\mathrm{AOD}_{total}$). First, the relative uncertainties of the direct horizontal irradiance ($\Delta\mathrm{DNI}_{total}$) is calculated from its individual contributions as follows:

$$\Delta\mathrm{DNI}_{total} = \sqrt{\Delta\mathrm{DNI}_{noi}^2 + \Delta\mathrm{DNI}_{mot}^2 + \Delta\mathrm{DNI}_{cal}^2}$$

The uncertainty of the motion correction ($\Delta\mathrm{DNI}_{mot}$) is taken from a pre-calculated lookup table, the calibration uncertainty ($\Delta\mathrm{DNI}_{cal}$) from the comparison of all calibration certificates, which turns out to be $\pm 2\%$ for all stable channels. The uncertainty caused by high-frequency fluctuations ($\Delta\mathrm{DNI}_{hf}$) is calculated during processing from the uncertainty of the fit parameters.

Table 1 summarizes the estimated uncertainty for the land campaign (Melpitz-Column) and ship borne cruise PS83. As mentioned in Sect. 2, the responsivity of some channels has been found to change significantly, and therefore been excluded from further uncertainty analysis. For the three channels (305, 340 and 380 nm), this issue should be fixed for ongoing measurements due to the modification of the instrument mentioned previously. All other channels show a uncertainty of 2.37% and 4.24% for a 95% CI, for the irradiance measurements on land and ship respectively.

$\Delta\mathrm{DNI}_{total}$ translates from Eq. (1) to $\Delta\mathrm{OD}_{\mathrm{DNI}}$ as follows:

$$\Delta\mathrm{OD}_{\mathrm{DNI}} = \frac{\mathrm{d}\tau}{\mathrm{d}\,\mathrm{DNI}} \Delta\mathrm{DNI}_{total} = -\mu_0 \frac{\Delta\mathrm{DNI}_{total}}{\mathrm{DNI}} \tag{15}$$

After all uncertainty components are calculated, the $\Delta\mathrm{AOD}_{total}$ sums up all components:

$$\Delta\mathrm{AOD}_{total} = \sqrt{\Delta\mathrm{OD}_{\mathrm{DNI}}^2 + \Delta\tau_R^2 + \Delta\tau_{gas}^2}$$

With the uncertainty of the Rayleigh-OD $\Delta\tau_R$ and $\Delta\tau_{gas}$ as the standard uncertainty for the OD from $O_3$, $NO_2$, $H_2O$, $CH_4$ and $CO_2$. Table 1 also shows the estimated uncertainty for the AOD calculations in absolute values for each stable channel. The AOD is calculated with a uncertainty of 2.65% for a 95% CI, for Melpitz-Column measurements, which is 0.0032 in absolute values for the 510 nm channel.

## 4.2 GUVis vs. Cimel observations

To verify our estimate of the uncertainty of the GUVis instrument as discussed in the previous section based on theoretical considerations, we have operated the instrument in close vicinity of an AERONET Cimel sun photometer during the Melpitz-Column campaign. AERONET sun photometers have a very strict calibration and quality assurance protocol, and are thus used as reference observations here. On land, when stabilization is not an issue, sun photometers are also the preferred method for aerosol characterization, due to the fact that the direct normal and not the direct horizontal irradiance is measured. Firstly, this results in a better signal to noise ratio particularly at low sun elevations. Secondly, the separation of irradiance components is





avoided, which introduces an additional uncertainty in the data analysis of shadow band radiometer measurements. Comparing both instruments is a good benchmark to test the reliability of the shadow band radiometer observations and the derived data products.

From matching channels, the observed spectral direct beam transmittance $T$ of the GUVis $T_G$ and Cimel $T_C$ instruments are

5 compared. The spectral direct beam transmittance has been calculated from Eq. (1) and Eq. (2). For $T_G$, the observed DNI, and for $T_C$ the retrieved total OD as reported by AERONET has been used to calculate the corresponding values of the direct beam transmittance. We decided to compare the transmittance rather than AOD, because this quantity is more directly related to the instrumental measurements. Specifically, the non-linearity introduced by the Beer-Lambert law and processing uncertainties in Rayleigh scattering and gas absorption are avoided.

A comparison for matching channels of both instruments is shown in Fig. 7, and corresponding regression parameters are listed in Table 1. The comparison shows a robust linear behaviour with increasing deviations for longer wavelengths. The slopes are close to the ideal value of unity for most channels, with a difference below 3%, except for the 443 nm and 510 nm channels, which exhibit deviations of about 3.4% and 5.7%, respectively.

To find out which of the instruments provides the more robust observations, we assessed the observations in terms of their

wavelength dependence. The resulting comparison is shown in Fig. 8. It presents the relative deviation of each matching channel of the GUVis and Cimel instruments from a regression line, which has been calculated assuming that the wavelength dependence can be modelled by the Ångström exponent. This regression has been calculated using a second order polynomial equation according to King and Byrne (1976):

$$\ln\left(\text{AOD}(\lambda)\right) = a \cdot \ln\left(\frac{\lambda}{\lambda_0}\right)^2 + b \cdot \ln\left(\frac{\lambda}{\lambda_0}\right) + c \tag{16}$$

Here, $a$ corresponds to the curvature in $\ln\left(\text{AOD}(\lambda)\right)$ versus $\ln\left(\frac{\lambda}{\lambda_0}\right)$ due to the departure of the aerosol size distribution from the Junge power law (Kaufman, 1993; Junge, 1955). Furthermore, $b$ corresponds to the Ångström exponent, and $c$ to the AOD at a reference wavelength $\lambda_0 = 500$ nm. This has been calculated using all observations during the Melpitz-Column experiment. We have restricted the comparison to matching channels with wavelengths of 870 nm and below, because a robust Ångström behaviour is only expected for these wavelengths for typical aerosol conditions. For both instruments, the AOD has

been calculated using our own algorithms for the estimation of gas absorption and Rayleigh ODs to ensure consistency. The comparison shows that the Cimel instrument provides an overall closer match to the regression line, as well as a lower scatter compared to the GUVis instrument. This behaviour suggests that both random and systematic uncertainties are lower for the Cimel observations, the latter likely attributable to a more accurate calibration. While it remains unclear at this stage to what extent it is possible to reduce the random uncertainty, the systematic deviations could be minimized by relying on a cross

calibration of matching channels with the Cimel instrument.

Figure 9 compares the Rayleigh scattering and gas absorption OD obtained from our scheme and the AERONET processing to identify differences in the retrievals. Here, our retrieval is applied for the central wavelengths given by the Cimel sun photometer, to concentrate on the inherent retrieval differences. The figure shows only small difference of gas absorption optical depths between both algorithms. While AERONET uses climatological means for the gas column density of ozone and





$NO_2$, we rely on satellite products from the AURA-OMI satellite instrument. The Rayleigh OD also shows a minor difference due to deviations of the used air pressure measurements.

Accepting these minor differences, the robust linear behaviour shown in Fig. 7 assures us that both instruments provide comparable products, and the deviation from the regression line $E_T$ can be translated from Eq. (1) into a measurement uncertainty for both the direct normal irradiance ($\Delta DNI$) and atmospheric OD ($\Delta OD$) of the GUVis, using the observations from the Cimel instrument as reference.

$$\Delta DNI = \frac{d\,DNI}{dT} \Delta T = \frac{I_0}{R_E^2} \Delta T \tag{17}$$

$$\Delta OD = \frac{d\tau}{dT} \Delta T = -\frac{\mu_0}{T} \Delta T \tag{18}$$

This uncertainty has been calculated for different situations in the atmosphere (e.g. only marine aerosol, desert dust or continental aerosol). Values for the typical AOD were chosen using the classification scheme from Toledano et al. (2007), and the uncertainty estimated using the standard deviation of the direct beam transmittance obtained from the comparison of both instruments. The uncertainties show a similar magnitude to that obtained from theoretic arguments in Sect. 4.1. It also shows that the uncertainty is strongly dependent on the observation conditions, specifically the aerosol loading and sun elevation.

## 4.3 GUVis vs. Microtops II observations

The German research vessel *Polarstern* is an ice breaker operated by the Alfred-Wegner Institute and mainly intended for Polar research. In autumn and spring of each year, transit cruises take place across the Atlantic ocean for transferring the ship into the corresponding polar summer hemisphere. Since 2007, these transit cruises are used to carry out atmospheric measurements within the framework of the OCEANET project (Macke, 2009). During the cruise PS83 in spring 2014, the GUVis shadow band radiometer was operated for the first time as part of OCEANET, with the aim of providing automated measurements of aerosol optical properties and its radiative effects. The track of this cruise is shown in Fig. 10.

Maritime aerosol consisting of sea-salt, sulphate and water was observed throughout the cruise. Continental influences were insignificant in the southern hemisphere, but became more prominent in the northern hemisphere. Mineral dust aerosol as well as biomass burning aerosol was observed while passing along the African coast West of the Saharan desert from 17th of March until 27th of March 2014.

The shadow band radiometer was installed on the navigation deck of the ship as far away as possible from ship superstructures to minimize shading effects. Only the mast and chimney as well as the smoke plume of the ship were able to shade the sensor under certain geometries and wind conditions.

Sun photometer observations with Microtops instruments were also taken during the cruise PS83 as a contribution to the MAN by scientists from the Max-Plank-Institute of Meteorology. These measurements were carried out manually every 10 to 15 minutes during clear sky conditions, and include five spectral channels ranging from 380 nm to 870 nm. The hand-held photometer is manually pointed towards the sun, taking a sequence of ten measurements. Before each measurement, the sky



condition is checked by eye to be cloud free, and to minimize the influence of the ship's smoke plume. Since the Microtops is a hand-held instrument, the smoke plume can be avoided by selecting another position on the ship for the measurement, in contrast to the fixed position of the GUVis instrument.

After quality control, the mean of these ten measurements are stored as final data set (Smirnov et al., 2002). The data is available from the website of the Goddard Space Flight Center of NASA and is used here as reference for the shadow band radiometer measurements.

The alignment information for the motion correction of the GUVis instrument are taken from the ship's marine inertial navigation system. This system provides precise measurements of the roll, pitch and heading angle with high temporal resolution. Detailed meteorological data are also available from the ship's weather station, and can be obtained from the DSHIP database system.

For quality assurance, quality flags were added to the observational data for different conditions. To investigate the influence of the smoke plume of the ship on the measurements, the relative wind speed and direction was used together with the sun position to determine the likelihood of the smoke plume passing between the sun and the shadow band radiometer sensor. Also, the deviation of the ship from horizontal due to the swell was used for a quality flag. Data with a misalignment of five degrees and higher are marked as high swell. Due to larger misalignments of the ship caused by higher swell, the uncertainty of the misalignment correction is expected to increase as described in Sect. 3.1.

Figure 11 shows the daily mean values of AOD obtained from the Microtops and GUVis measurements during the whole cruise. Shown is also the uncertainty estimate as described in Sect. 4.1. The GUVis time series has been filtered to only include data points which occurred within five minutes of a Microtops measurement. The curves obtained from both instruments agree very well. The time series shows low AOD for most of the cruise. An increase of AOD is evident while passing the Sahara desert and close to the European continent at the end of the cruise.

This behaviour can also be seen in Fig. 12, where Microtops and GUVis measurements agree well classified into different aerosol types following the method of Toledano et al. (2007). Marine aerosol dominates throughout the cruise as expected. However, desert dust can clearly be identified while passing the Sahara as shown in Fig. 10. At the end of the cruise, the influence on the continental aerosol type increases.

The comparison shown in Fig. 13, as well as regression parameters quoted in Table 1 show an overall agreement of the spectral direct beam transmittance observations from both instruments with a deviation below 4%, which is in the same range as the comparison to the Cimel sun photometer. This finding highlights the suitability of the GUVis instrument for ship borne operation. For the comparison, only non flagged data have been considered (e.g. at low swell and no smoke plume over the instrument). In principle, we do expect an increase of the uncertainty with increasing swell, however we have been unable to identify this based on the current data likely due to the limited number of observations with high swell conditions. In contrast, the influence of the smoke plume can clearly be identified in the comparison, with the smoke flag reliably excluding outliers from the whole dataset.

We plan to continue the investigation of the instrumental uncertainty with additional observations from ship cruises in the future to better quantify the effects of swell and different aerosol types on the observational accuracy.



## 4.4  Discussion of the uncertainty

Considering the different sources of uncertainty, it turns out that the calibration uncertainty is the dominating contribution to the total measurement uncertainty of the GUVis instrument. Here, we have assumed that the calibration uncertainty is equal to the temporal change between two laboratory calibrations separated by 2 years. This change is found to be less than 2% for

most channels, but can reach up to 40% for the channels with soft-coated filters (e.g. the 750, 940, and 1550 nm channels).

From the 940 nm channel, the precipitable water column amount can be inferred with an uncertainty of $\pm0.034$ cm as is demonstrated in Sect. 3.3, if the calibration is well-known. Currently however, the accuracy is limited by the temporal stability of the soft-coated filter used for this channel. While the exchange of the filter with a hard-coated one would be the best solution, frequent inter-calibration based on parallel observations with an AERONET Cimel sun photometer and the methods presented

here can also ensure a high level of accuracy.

The channels below 380 nm were also found to have an abnormally high temporal drift. This issue has been attributed to a change in transmission of an insert below the diffuser of the instrument, which has been replaced by the manufacturer with a new material to overcome this issue.

The slight misalignment of two degrees during set up on land results in an motion correction uncertainty of 0.35% with a

15 95% CI for observations during the Melpitz-Column campaign. This emphasizes that a careful alignment of the instrument is essential to minimize this uncertainty. High frequency fluctuations cause an uncertainty of 0.56% with a 95% CI. The uncertainty on land is estimated to be 2.37% with a 95% CI for the stable channels.

This magnitude is confirmed by our comparison with observations from a Cimel sun photometer during the Melpitz-Column campaign (see Sect. 4.2). As the measurement principle of a sun photometer is more direct than the shadow band method of

20 the GUVis instrument, higher accuracy is expected, which is indeed confirmed by our results in Sect. 4.2. Nevertheless, the agreement of matching channels for both instruments is generally within 3%, corresponding to a standard deviation below 0.02 in direct beam transmittance, illustrating that the GUVis shadow band radiometer can compete with sun photometer measurements. Some questions remain open however for the uncertainty of the 443 nm and the 510 nm channels, which show comparatively large deviations of 3.4% and 5.7%, respectively. This uncertainty may result from the fact, that the GUVis is

25 calibrated using lamp calibrations and not with the Langley technique which is used to calibrate Cimel sun photometers.

If differences in the wavelengths of channels are corrected for, only minor deviations in the AOD retrievals based on the AERONET algorithms and our analysis have been found. These deviations result from the different methods of calculating the Ozone and $NO_2$ absorption optical depths.

The GUVis is well-suited for ship borne observation. Measurements on the ship are however additionally influenced by

30 the swell and are expected to exhibit a higher uncertainty than those on land due additional uncertainties introduced by the extrapolation and motion correction steps. Our estimate of the uncertainty for ship borne measurements of the direct beam transmittance is 4.24% with a 95% CI, which is in excellent agreement with the comparison to Microtops observations during the *Polarstern* cruise in spring 2014. Here, deviations up to 4% have been found for matching channels, and standard deviation up to 0.026 which is slightly higher than that found in the comparison with Cimel observations. It has to be noted, however, that



we also expect the Microtops sun photometer observations to be less accurate than the Cimel ones due to manually pointing the instrument at the sun on a ship.

At this stage we were not able to reliably determine the influence of the swell on the observational accuracy. This is mainly due to the limited amount of data available so far, in particular with higher swell due to the relatively calm sea conditions during the cruise PS83. We plan to revisit this point in the future, when observations from more cruises are available.

During ship borne operation, the instrument's 2-axis internal accelerometer is not sufficient to determine its position and alignment. While highly accurate systems such as *Polarstern's* navigation system measure the ship motion on most research vessels, an offset between the instrument and the ship's sensors due to an imperfect alignment can introduce additional uncertainty. Hence, an upgrade of the instrument with a sensor capable of measuring its position also in dynamically moving environments would further improve its usability for ship borne operation.

The calculation of the AOD from the direct beam transmittance is affected by an uncertainty of $2.65\%$ with a $95\%$ CI for stable channels , which is an absolute uncertainty of $0.0032$ with a $95\%$ CI for the $510\,\mathrm{nm}$ channel for Melpitz-Column measurements. The retrievals of AERONET and GUVis on AOD match closely as presented in Fig. 9, with minor differences caused by the different treatments of Ozone and $NO_2$ absorption. Also the direct comparison of the AOD retrieval with adjusted wavelengths show only small deviations in lower wavelength channels due to different methods of deriving Ozone- and $NO_2$-OD. As expected sun photometry is more accurate on a land site, but the GUVis can compete with this observations.

Our uncertainty estimate and the comparison with sun photometer observation presented here demonstrate that the GUVis shadow band radiometer is a reliable instrument for the observation of spectral irradiance components and aerosol properties both on land and on ships. For the latter, the automatic nature of its observations are a clear advantage over the Microtops instrument employed by MAN, which requires a human operator. The time series from the GUVis instrument is thus more continuous and has a higher time resolution than the time series of the Microtops. Nevertheless, one should be aware that in contrast to a human operator, the GUVis is mounted in a stationary position, and thus cannot avoid shadows from the ship super structure or the smoke plume. Hence, careful data analysis and quality screening of the raw data is essential to ensure high accuracy.

## 5 Conclusions and Outlook

The 19 channel shadow band radiometer GUVis was operated for the first time on the research vessel Polarstern during its cruise PS83, with the aim of providing automated measurements on the radiative effects and optical properties of aerosol as part of the OCEANET project (Macke, 2009). Due to its continuously moving shadow band, this instrument allows to determine the direct, diffuse and global components of the solar irradiance on a moving platform with high accuracy.

In this paper, the data analysis implemented at TROPOS is described, including algorithms for cloud masking, motion and cosine error correction, the separation of the different irradiance components, and the calculation of direct sun products. These methods are based to a large extent on Morrow et al. (2010) and Alexandrov et al. (2002), and have been adapted for application





to the GUVis instrument. The calculation of spectral AOD accounts for contributions by Rayleigh scattering and gas absorption to the total atmospheric OD, and uses satellite products for obtaining the column concentrations of $O_3$ and $NO_2$.

Our results confirm that the GUVis instrument can provide automated and accurate measurements of the spectral irradiance components and the optical properties and radiative effects of aerosol on ships. Especially the observation of all three spectral
radiation components simultaneously with one sensor is an advantage in comparison to sun photometers. Due to its stationary position, however, the influence of the ship exhaust needs to be taken into account. More observations are also required to assess the long time stability and the uncertainty under high-swell conditions.

Some questions still remain concerning filter stability, calibration accuracy and the overall retrieval performance, which we plan to investigate in future work. In the next years, the GUVis instrument will be routinely operated as part of the TROPOS
OCEANET container on RV *Polarstern* to carry out measurements of spectral irradiances and AOD, and to investigate the solar radiation budget over the Atlantic ocean. Regular calibrations of the instrument are planned to ensure the stability and overall performance of the instrument. Here, cross calibration with a AERONET Cimel sun photometer on land constitute an accurate alternative to laboratory calibrations, if only for the channels also available from the AERONET instruments.

Besides the current set of products, we are going to investigate the potential for further aerosol products such as the single
scattering albedo and asymmetry parameter by using the diffuse to direct ratio as outlined by Herman et al. (1975) and applied in a number of aerosol studies (e.g., Petters et al., 2003; Kassianov et al., 2007). The GUVis is very well suited for this use, because the diffuse and direct irradiance are measured simultaneously with only one sensor, causing negligible calibration uncertainty.

A synergistic analysis also utilizing images from the all sky camera will allow an improved detection of clouds (Heinle
et al., 2010). Specifically, this can help to improve the identification of short periods with cloud gaps, thereby enhancing the interpretation in broken cloud conditions and improving the separation of cloud and aerosol radiative effects. Targeting clouds, an adaptation of the retrieval methods presented by Brückner et al. (2014) and Min and Harrison (1996) could be applied to estimate cloud properties from the GUVis measurements either stand-alone or in synergy with microwave radiometer observations. Finally, super site observations including active instruments such as cloud radar and lidar could be used to extend
previous efforts directed at testing radiation closure studies (e.g., Ebell et al., 2011) to narrowband irradiance observations.

*Acknowledgements.* We thank Patric Seifert for his effort in establishing and maintaining the AERONET measurements at the Melpitz site during the Melpitz Column experiment. We thank Stefan Kinne and Alexander Smirnov for their effort in maintaining and organisation of Microtops observations on RV *Polarstern*, and Dagmar Popke and Gaby Rädel for operating the Microtops during the cruise PS83. Thanks are also due to the Alfred Wegener Institute for Polar and Marine Research (AWI) for the opportunity to operate the GUVis instrument during
the research cruise PS83 across the Atlantic Ocean on RV *Polarstern*.





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



**Table 1.** Summary of the main results of our evaluation of the GUVis shadow band radiometer. Sorted by wavelength, the relative change in calibration of each channel is shown in column two. Channels with soft-coated filters (750 nm, 940 nm, 1550 nm), and channels affected by a change in transmission of a diffuser insert (305 nm, 340 nm, 380 nm) are excluded from the uncertainty estimation. The mean uncertainty and deviation according to the 95% CI from our analysis (Sect. 4.1) are shown for irradiances for all stable channels separately for land based and ship borne observations. The mean uncertainty and deviation of 95% CI for the calculation of AOD is shown in absolute uncertainties in the next column, which is between 2.01% to 2.66% for all channels. The linear regression parameters obtained from the comparison of GUVis with Cimel and Microtops spectral direct beam transmittance observations are given in the final 6 columns.

| Channel | Calibration deviation | Uncertainty land | ocean | Aerosol | Comparison to Cimel slope | $\sigma$ | R | Comparison to Microtops slope | $\sigma$ | R |
|---|---|---|---|---|---|---|---|---|---|---|
| [nm] | [%] | [%] | [%] | $[10^{-3}]$ | [-] | [-] | [-] | [-] | [-] | [-] |
| 305 | 27.98 | - | - | - | - | - | - | - | - | - |
| 340 | 10.86 | - | - | - | 1.019 | 0.006 | 0.998 | - | - | - |
| 380 | 2.16 | - | - | - | 1.003 | 0.008 | 0.998 | 1.026 | 0.029 | 0.971 |
| 412 | -0.56 | $2.30 \pm 0.07$ | $3.50 \pm 0.40$ | $4.81 \pm 0.148$ | - | - | - | - | - | - |
| 443 | -2.02 | $2.23 \pm 0.02$ | $3.48 \pm 0.40$ | $4.45 \pm 0.145$ | 0.966 | 0.010 | 0.997 | 1.004 | 0.024 | 0.967 |
| 510 | -0.55 | $2.20 \pm 0.02$ | $3.46 \pm 0.37$ | $3.09 \pm 0.135$ | 1.057 | 0.013 | 0.994 | 1.040 | 0.028 | 0.975 |
| 610 | -0.73 | $2.20 \pm 0.03$ | $3.47 \pm 0.35$ | $3.10 \pm 0.128$ | - | - | - | - | - | - |
| 625 | -0.68 | $2.21 \pm 0.03$ | $3.48 \pm 0.36$ | $2.93 \pm 0.130$ | - | - | - | - | - | - |
| 665 | -0.60 | $2.21 \pm 0.03$ | $3.48 \pm 0.35$ | $2.17 \pm 0.129$ | 1.028 | 0.015 | 0.987 | 1.029 | 0.026 | 0.958 |
| 694 | -0.13 | $2.21 \pm 0.04$ | $3.59 \pm 0.40$ | $3.35 \pm 0.124$ | - | - | - | - | - | - |
| 750 | 18.40 | - | - | - | - | - | - | - | - | - |
| 765 | -1.40 | $2.20 \pm 0.03$ | $3.65 \pm 0.45$ | $8.71 \pm 0.134$ | - | - | - | - | - | - |
| 875 | -1.55 | $2.24 \pm 0.04$ | $3.65 \pm 0.45$ | $2.11 \pm 0.136$ | 1.014 | 0.019 | 0.961 | 0.987 | 0.026 | 0.974 |
| 940 | -9.18 | - | - | - | - | - | - | - | - | - |
| 1020 | -1.24 | $2.25 \pm 0.05$ | $3.64 \pm 0.42$ | $2.27 \pm 0.140$ | 1.002 | 0.015 | 0.965 | - | - | - |
| 1245 | -0.36 | $2.24 \pm 0.04$ | $3.58 \pm 0.39$ | $1.68 \pm 0.142$ | - | - | - | - | - | - |
| 1550 | -40.43 | - | - | - | - | - | - | - | - | - |
| 1640 | -0.73 | $2.29 \pm 0.06$ | $3.77 \pm 0.47$ | $1.99 \pm 0.142$ | 1.013 | 0.018 | 0.922 | - | - | - |





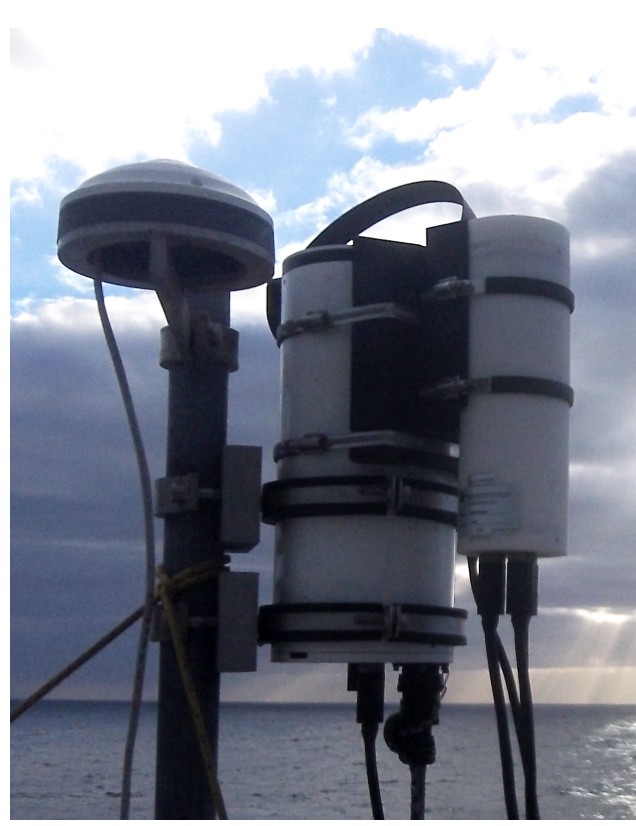

**Figure 1.** The shadow band radiometer GUVis mounted on the research vessel *Polarstern* during the cruise PS83 together with a total-sky imager.





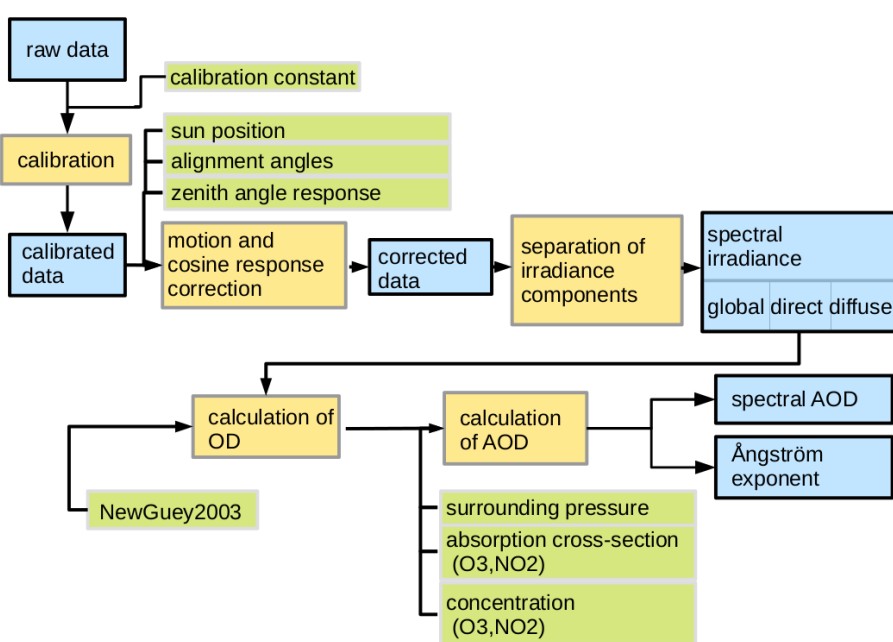

**Figure 2.** This flowchart outlines the data processing steps for the GUVis observations. Generated data products are shown in blue, while calculation and processing steps are shown in yellow, and supplementary data needed for processing are shown in green.



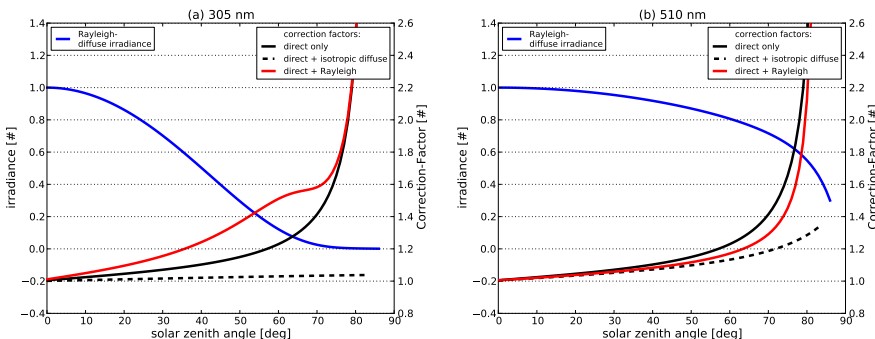

**Figure 3.** This figure shows correction factors for the motion correction for measurements of the 305 nm and the 510 nm channel of the GUVis shadow band radiometer. The two panels show the calculated diffuse irradiance (blue), which is normalized to its maximum. Additionally the panels show three correction factors calculated for an inclination of 6° of the ship towards the sun's azimuth angle (e.g. high swell). The direct only (black solid) correction factor refers to $C1$ described by Bannehr and Schwiesow (1993). By adapting the method of Boers et al. (1998) and using radiative transfer calculations carried out with the libradtran package using the DISORT solver (Mayer and Kylling, 2005), spectral correction factors $C2$ (black dashed) and $C3$ (red) are calculated taking direct and diffuse irradiance into account.





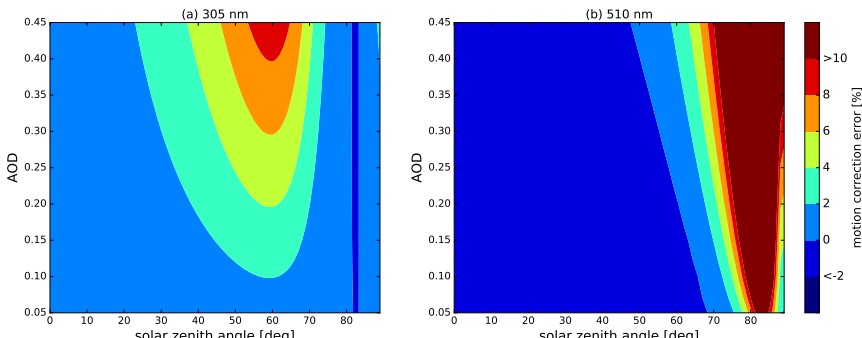

**Figure 4.** This figure demonstrates the error of the motion correction for measurements of the 305 nm and the 510 nm channel of the GUVis shadow band radiometer. The error is calculated comparing $C3$ with neglected aerosol (AOD = 0), to correction factors with additional aerosol influence. These correction factors are calculated like $C3$ but using radiative transfer calculations with aerosol type and properties according to Shettle (1990) with AOD's of 0.05 to 0.45. The calculations are done for an inclination of $6°$ of the ship towards the sun's azimuth angle(e.g. high swell).



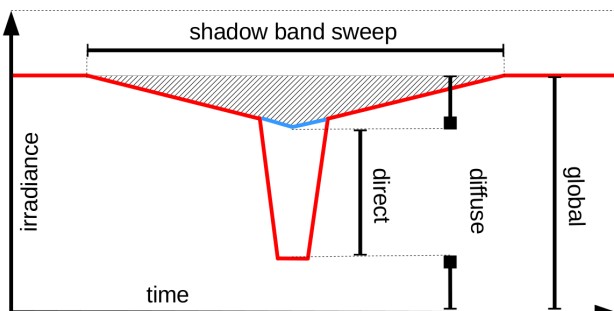

**Figure 5.** Idealized irradiance time series measured during one shadow band sweep. When the sun is blocked, some part of the diffuse irradiance (black hatched area) is blocked by the shadow band in addition to the direct sun light. This part is estimated by extrapolation of the data from the time series (blue line). From data obtained during the sweep, the direct and diffuse irradiance is calculated. Between the sweeps, the global irradiance is observed.



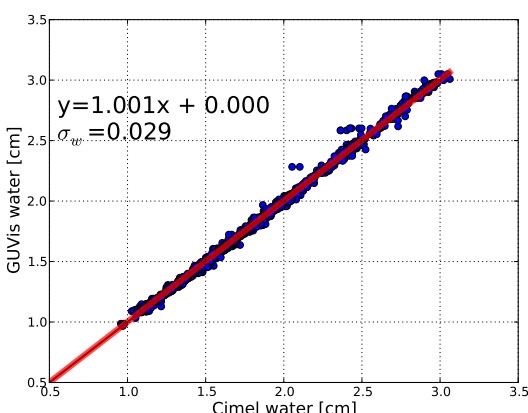

**Figure 6.** Relation of the precipitable water obtained from the Cimel sun photometer and the GUVis shadow band radiometer during Melpitz-Column experiment.





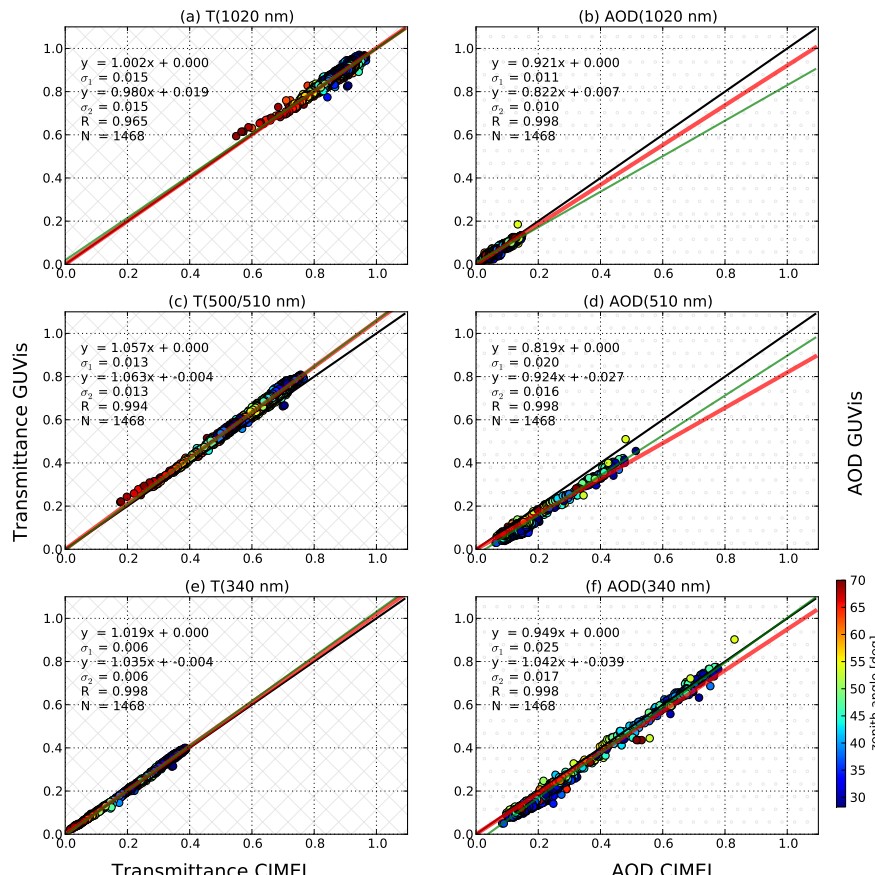

**Figure 7.** Comparison of the spectral direct beam transmittance and spectral AOD for three matching channels of the instruments GUVis and Cimel sun photometer. The parameters of the linear regressions with free intercept and intercept forced to zero are denoted in each panel. Also the deviation from the regression lines are denoted as $\sigma_1$ and $\sigma_2$. R denotes the correlation coefficient and N the number of available measurement points for comparison. The points are colored with respect to the zenith angle.





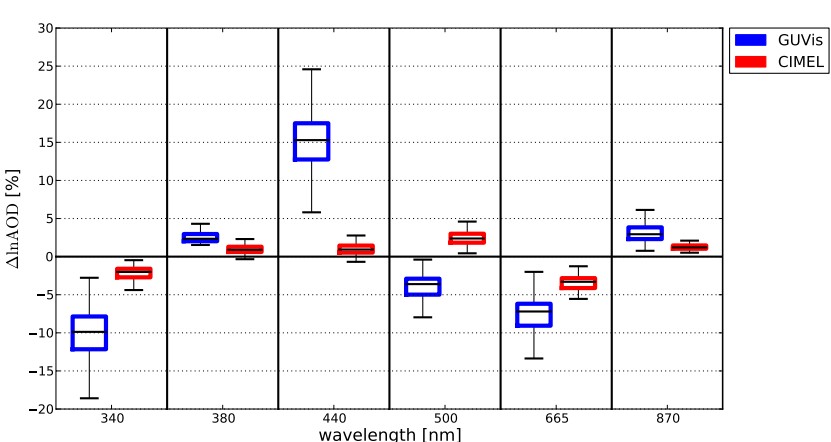

**Figure 8.** Deviation of matching channels of the Cimel and GUVis instruments from the Ångström behaviour as obtained from a linear regression, displayed in a box and whisker plot. Shown are the median, boxes extending to the 25th percentile, and whiskers extending to the 75th percentile of the data.





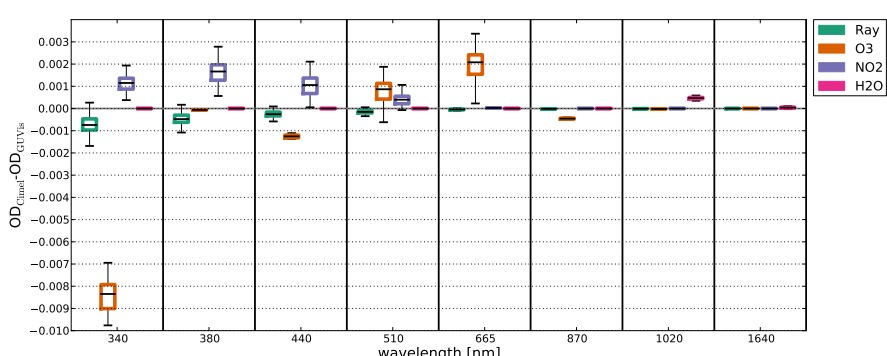

**Figure 9.** Comparison of the retrieved mean OD and its components for matching channels during the Melpitz-Column campaign. Shown is the difference of the Cimel and the GUVis retrieved OD components in a box and whisker plot. The median is displayed, the box extend to the 25th percentile and the whiskers towards the 75th percentile of the data. Shown are the OD for Rayleigh (Ray), ozone ($O_3$), nitrogen dioxide ($NO_2$) and water vapor ($H_2O$). To highlight the differences in the retrieval scheme, we have adjusted the central wavelengths of the GUVis channels to those of the Cimel instrument.





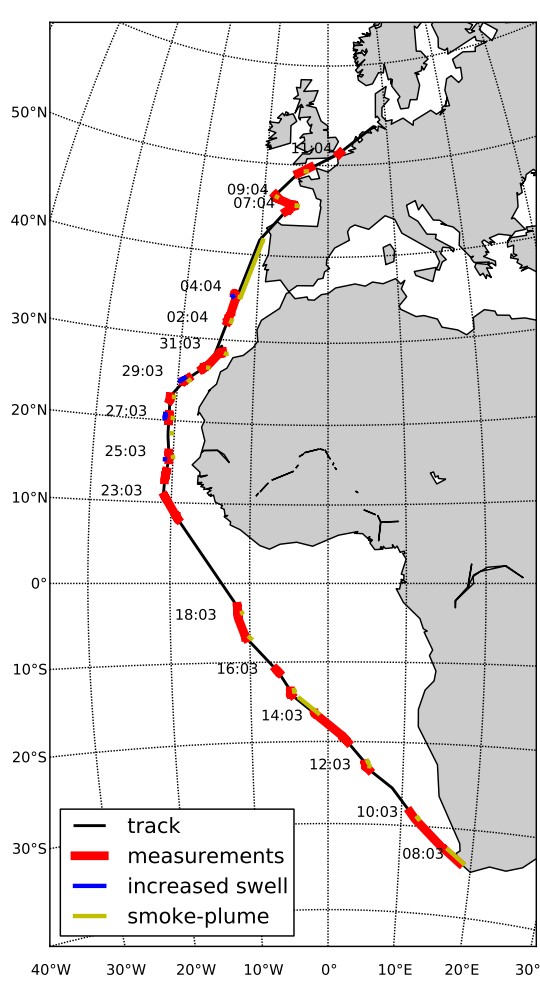

**Figure 10.** The track for cruise PS83 for the research vessel *Polarstern*. Track points with observations available from both the GUVis and Microtops are marked in red. Additionally high swell conditions during the cruise are marked blue, and the possible influence of the ship's smoke plume on GUVis observations is marked in yellow.



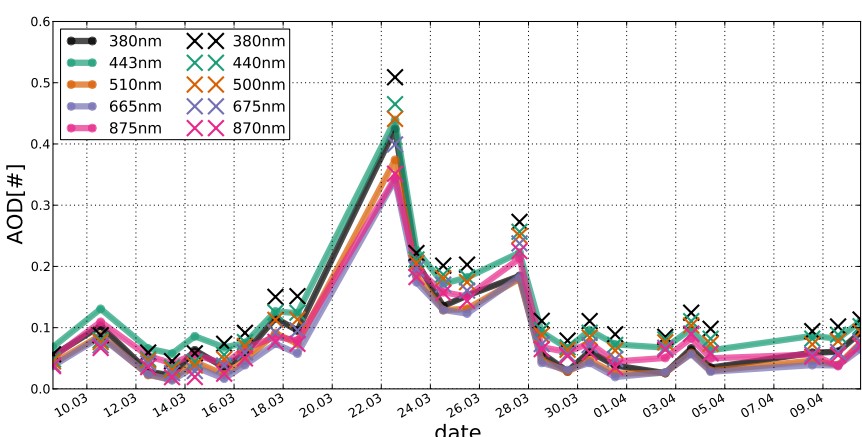

**Figure 11.** Daily mean values of AOD from the Microtops (cross) and GUVis (dot) observations. Observations of the GUVis within five minutes to the Microtops observations are considered only.





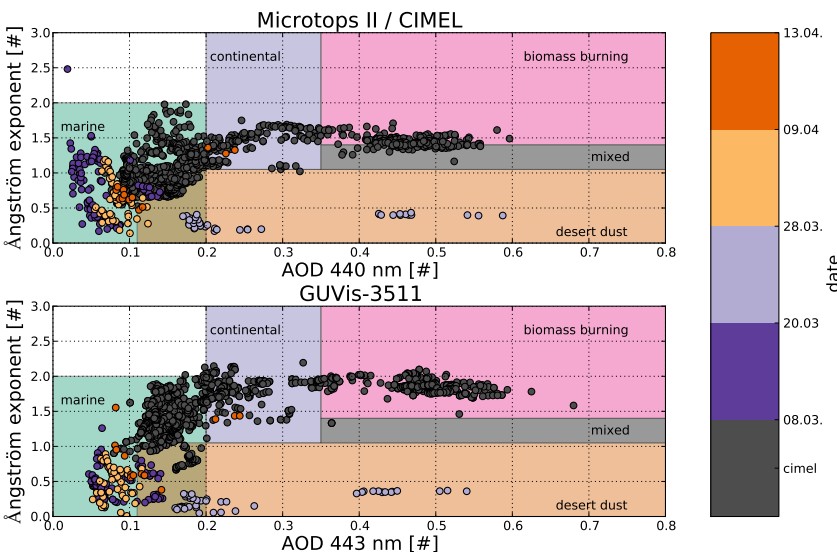

**Figure 12.** Comparison of Microtops (colored), Cimel sunphotometer (grey) and GUVis time series using the Ångström exponent (440 nm - 870 nm) and the AOD at 440 nm for aerosol classification as described by Toledano et al. (2007). For the ship measurements, the data points are colored according to their date, as referenced by the colorbar.





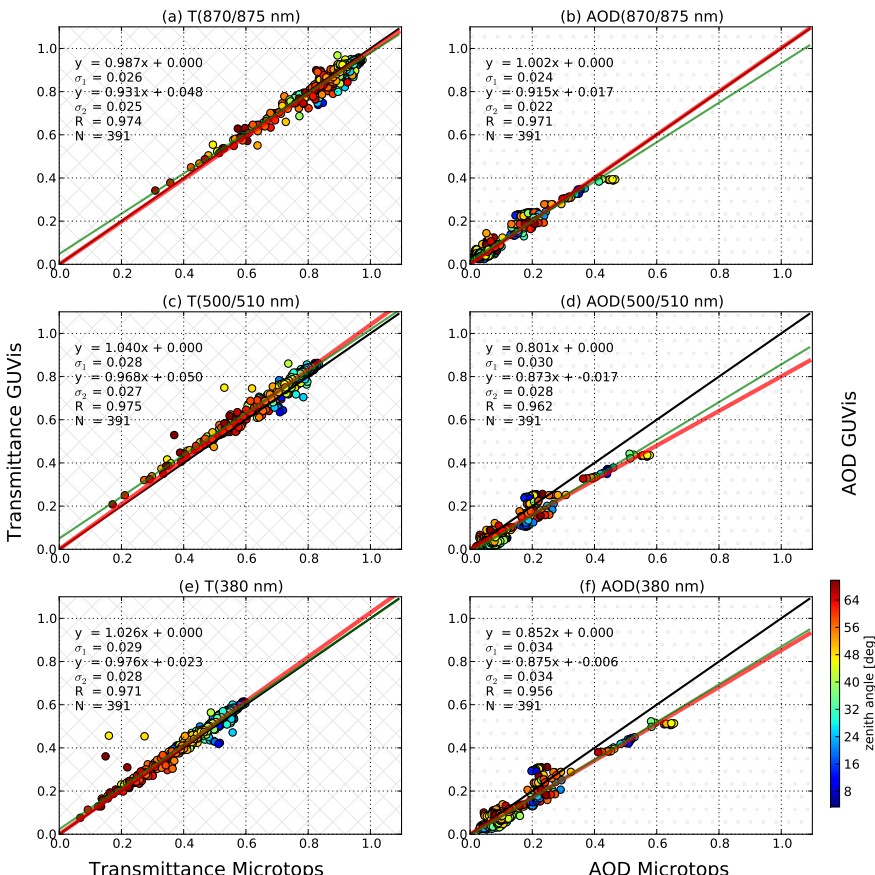

**Figure 13.** Comparison of the spectral direct transmittance and spectral AOD for three matching channels of GUVis and Microtops sun photometer. The parameters of the linear regressions with free intercept and intercept forced to zero are denoted in each panel. Also the deviation from the regression lines are denoted as $\sigma_1$ and $\sigma_2$. R denotes the correlation coefficient and N the number of available measurement points for comparison. The points are colored with respect to the zenith angle.