# Peer review of "Ship borne rotating shadow band radiometer observations for the determination of multi spectral irradiance components and direct sun products for aerosol"

_Atmospheric Measurement Techniques, 2016_

## Referee Comment (RC1) · Anonymous Referee #1 · 2 Nov 2016

There is much to like in the submitted publication- Ship borne rotating shadow band radiometer observations for the determination of multi spectral irradiance components and direct sun products for aerosol written by J. Witthuhn, H. Deneke, A. Make and G. Bernhard. I cannot, however, recommend that it be published at this time because I think it is incomplete. The authors make the assumption that the solar spectral irradiance at the top of the atmosphere is perfectly known. This is not true and this source of error needs to be taken into account in their analysis as was the case in Miller et al, 2004. Furthermore this new device needs a Langley calibration. I strongly disliked figure 12. I recommend keeping the land based measurements in one part of the fig-

ure and the marine observations in another. I would also include designated regions for different aerosol types as was done. In section A of the figure plot the Cimel and GUVis data and in section B plot the Microtops and GUVis data. I found a discrepancy between the text and figure for figure 11. "Figure 11 shows the daily mean values of AOD obtained from the Microtops and GUVis measurements during the whole cruise. Shown also is the uncertainty estimate as described in Sect.4.1." I don't believe the uncertainty is shown. I would like to see the figure redrawn. I think it would be better without the lines and with points with errorbars for the GUVis data that can be clearly seen. A separate portion should be used to show the difference between Microtops and GUVis mean observations.

Miller, M.A., M.J. Bartholomew, and R.M. Reynolds, 2004. The accuracy of marine shadow-band sun photometer measurements of aerosol optical thickness and Ångström exponent, Journal of Atmospheric and Oceanic Technology, v21, p. 397 – 410.

---

## Referee Comment (RC2) · Anonymous Referee #2 · 9 Nov 2016

This manuscript by Witthuhn et al., "Ship borne rotating shadow band radiometer observations for the determination of multi spectral irradiance components and direct sun products for aerosol", describes an instrument (GUVis-3511) suitable for ship-borne measurements of direct, diffuse and global spectral irradiance and of derived products, e.g. aerosol optical depth. I find the manuscript interesting and relevant to introduce an instrument suitable for these measurements in demanding conditions. This topic is also suitable to the scope of AMT. Much of the manuscript is devoted to the discussion of the uncertainties and that is how it certainly should be in this case. However, somehow this part does not convince, and the reader is left somehow confused or at least I

was. I think this discussion should be more thorough and coherent.

Main comments:

Just couple of examples about those places in the text that contributed to this confusion regarding the uncertainties. In Page 18, Line 6 you said that ".. causing negligible calibration uncertainty" and in the page 15, line 24 you said that "... the calibration uncertainty is the dominating contribution ...". So the dominating type of uncertainty is negligible? Is the accuracy indeed so perfect? You also compared to AERONET and said "that both random and systematic uncertainties are lower for the Cimel observations .." than for GUVis-3511. AERONET AOD has a typical uncertainty of 0.01–0.02 (Eck et al., 1999) and you give an estimate of 0.0032 at 510nm. I understand that you give a relative uncertainty and AERONET uncertainty is an absolute estimate, still it seemed that there was a some sort of inconsistency or contradiction here and your uncertainty estimate "too good to be true"? The discussion about uncertainties should be more thorough and clear.

You did not explicitly discuss the fraction of aureole irradiance that gets blocked as well or is it so that your 2.5cm band does not introduce a significant blocking angle?

Minor comments:

Page 2, line 18: "... measurements of aerosol optical properties and radiative fluxes ..." Confusing sentence, when the instrument measures radiative fluxes only and aerosol optical properties are derived.

Page 5, line 16: Acronym OD, optical depth, was never introduced.

This is likely not to be included in the scientific paper, but for my own curiosity: what is the price of this instrument?

Eck, T. F., Holben, B. N., Reid, J. S., Dubovik, O., Smirnov, A., O'Neill, N. T., Slutsker, I., and Kinne, S.: Wavelength dependence of the optical depth of biomass burning, urban, and desert dust aerosols, J. Geophys. Res., 104, 31333–31349, 1999.

---

## Referee Comment (RC3) · Anonymous Referee #3 · 9 Nov 2016

General comments

The paper describes a shadowband radiometer, which has been developed to measure AOD and global, direct and diffuse irradiances at 19 wavelengths in the UV, visible and shortwave IR. The advantage of this instrument is that it can be used for automated measurements on a ship. In principle, this is an important extension of the current measurement capability for atmospheric radiation and aerosol measurements during ship cruises.

The paper describes the instrument, its calibration, the AOD retrieval algorithm, and an

error analysis of the irradiance and AOD measurements.

a. In general, the paper is (in parts) well readable, and the algorithm steps and errors are described in detail. However, the structure is often confusing and not logical. Different topics are mixed in one section. Reordering of sections and figures is needed. The overall presentation should be clearly improved. See specific comments below.

b. The theoretical AOD error estimate of Sect. 3 should be confronted with the real AOD errors of Sect. 4, preferably in a separate discussion section for land and ocean. Now the Melpitz field campaign over land gets much more attention than the Polarstern field experiment, whereas the latter is the real innovative application.

c. The paper is missing a description of the advantage/purpose of measuring at 19 wavelengths. Only results for a few wavelengths are shown. Please add more spectral results to show the capability of the instrument for future applications.

The paper could be published when the comments are taken into account, especially a better structuring of the paper.

Specific comments

1. The title should include the terms "algorithm" and "error analysis". Please replace the vague term "direct sun products" by AOD.

2. Abstract: define CI. This acronym is used throughout the paper without any explanation. Please replace where possible by a clear term.

3. Throughout the paper errors are given as percentages with 2 decimals. This suggests an accuracy that is not attainable, as is shown by the results. For example, the abstract mentions 4.24 % total uncertainty. See also Table 1. Please reduce the number of decimals to 1 or 0.

4. How to calibrate in the field the fast degrading 750 nm and 1550 nm channels? Could you use a fixed relation between stable and unstable channels for specific

scenes ?

5. Sect. 3 contains three topics: the correction steps, the AOD retrieval algorithm, and the error analysis. Please separate these three topics in in three sections.

6. Equation 1: define \tau. In general: define all symbols directly when they are used.

7. Equation 1: it is strange that you define R_E as a ratio of distances and not as a distance. Please use a more appropriate symbol.

8. Add directly below Equation 1 in an equation that you assume m = 1/mu0.

9. P. 5, l. 21: "sun below a zenith angle of 70 deg" : this is unclear, please rephrase (occurs more often)

10. P. 5, l. 28: first step: how are the steps numbered?

11. Equations 3-5: How are these factors C1, C2, C3 used in your algorithm? What is the correction formula?

12. P. 6, l. 18, l. 24: The deviation of what from what?

13. P. 6, l. 18: lower wavelengths > smaller wavelengths

14. P. 7, l. 12-13: repetition of text. Refer to the above subsection.

15. P. 7, l. 21: Please give the resulting error that follows from Fig. 4.

16. P. 8, l. 28: ... is calculated as the difference between the global and the direct irradiance.

17. P. 9, l. 20: please start a new subsection here on H2O channel calibration.

18. Equation 10: please remove the superfluent term X

19. P. 9, l. 27: transmittance from Rayleigh ... > extinction by Rayleigh . . .

20. P. 10, l. 1-14: this part is very unclear. Please remove if possible, since the use of

this channel is so debatable. Basically: what is the use of the GUVIs 940 nm channel which drifting so much that you need an alongside Aeronet measurement?

21. P. 10, l. 15 – l. 27: please move this part up, above the H2O discussion, since Rayleigh and ozone-NO2 correction is much more important than H2O correction.

22. P. 10. L. 16: uncertainty > absolute uncertainty \Delta

23. P. 10., l. 20: variate > varying

24. P. 10, l. 25: please give recent references on the accuracy of the current OMI product versions.

25. P. 11, l. 4: \tau_w > \Delta \tau_w

26. P. 10: Please summarize all OD errors from Sect. 3 in a Table.

27. Sect. 4 contains four different topics in one section: a theoretical uncertainty estimate in Sect. 4.1, two field experiments - one on land and one on ocean - and a discussion. Please detach these parts. Sect. 4.1 clearly belongs to the last part of Sec. 3, the theoretical error estimate. The two fields experiments, showing the real errors, are different in content, plots etc., and could be separated. The ship based measurements show the realistic capability of the instrument. The discussion in Sect. 4.4 deserves a separate discussion section in which the theoretical errors should be confronted with the real errors.

28. Please show also the differences in AOD between GUVis and Microtops in Fig. 11.

29. P. 12, sect. 4.1: Number the equations on l. 6 and l. 19.

30. P. 12, l. 6: What is DNI_noi ? Same as DNI_hf?

31. P. 12, sect. 4.1: Please do not use acronyms in equations but symbols. So please use I instead of DNI in the equation on l. 6 and eq. (15). Please use \tau instead of AOD and OD in Eq. on l. 19.

32. P. 13, l. 4: What are T_G and T_C?

33. P.13, l. 4-9: this paragraph is unclear, please rephrase.

34. P. 13, l. 33: only small. . .: the difference for ozone is very large.

35. P. 14, l. 4: what is E_T? it is not used in any equation.

36. P. 14, Eqs. 17 and 18: please give these equations earlier, in sect. 4.1, as part of (new) sect. 3 error analysis.

37. P. 15, l. 17-21: please give a quantitative result of the real GUVis AOD error on the ship, from comparison with the Microtops.

38. P. 17, l. 11 ff: Does the percentage error mentioned here relate to AOD? Percentage errors are not very useful for AOD, since the AOD is very variable. Only absolute errors are useful, which can also be seen from Eq. 1, which is the relationship between AOD and transmittance. This point holds for the entire AOD error discussion.

39. P. 18, l. 4: remove: and radiative effects (since this is not shown).

40. P. 18, l. 13: if only > but only

41. P. 18, l. 21-25: For these applications of this instrument, it should be demonstrated that the other wavelengths of the GUVis, for which no results were shown in this paper, are indeed functioning as required.

42. Table 1: Please clarify caption and header. Caption: please always indicate the number of the column. Header: Deviation of what? Uncertainty in what? What do slope, \sigma and R mean? Comparison to Cimel = Land? Comparison to Microtops = Ocean? Aerosol > AOD ?

43. Please add a table (or a column in Table 1) with the spectral bandwidth and central wavelength of each channel. For which wavelength was the OD calculated?

44. What is the shape of the spectral response functions of the 19 channels?

45. Figure 1: please explain what is what, e.g. with arrows. It would be helpful to have sketch of the GUVis, or a top view.

46. Figure 2: Please number the steps of the data processing algorithm. Calibration: radiometric or spectral? Surrounding pressure> surface pressure. Concentration > column density.

47. Figure 3: explain the two y-axes. The last lines of the caption are a repetition from the main text. Explain in the main text how C_i are used.

48. Figure 4: Caption: Error due to aerosols . . ..

49. Figure 5: This figure should be shown earlier, because it nicely shows the principle of the shadowband measurement.

50. Figure 6: this figure can be removed, since its content can be well described in a few words in the main text.

51. Figure 7: please explain the symbols of the legend in the caption and identify the two equations.

52. Figure 8: Caption: Deviation > Difference . . . Mention the Melpitz campaign and the time period.

53. Figure 9: Explain y axis: Difference in OD . . . Please zoom-in by removing the single outlier (mention specifically) and rescaling the y-axis.

54. Fig. 11: Label the GUVis and Microtops points. Add error bars. Show also the AOD differences GUVis – Microtops.

55. Figure 12: This is a very difficult and confusing plot. There is too much information. Land and ocean data are mixed? Are there also two color codings mixed? Please make separate figures. It anyway requires more explanation in the caption. Give also the year.

56. Fig. 13: This figure should be should before Figure 11, of course. Please give the dates and location in the caption.

57. Why is the slope of the transmittances in Fig. 13 closer to 1 than the AOD? OD correction differences?

Technical corrections

- Explain all acronyms the first time they appear: TROPOS, OCEANET, BSI, OD, . . .

- Explain all symbols the first time they appear. E.g. $T\_G$, $T\_C$, $E\_T$, . . .. are not explained.

- Write much used scientific and technical terms not as separate words, but as connected words: shipborne, shadowband, multispectral, subproject, airplane, etc.

- All symbols, either in text or in equations, should be in italics. For example, $T$ and $w$ in equation 9.

- All Acronyms should be in upright font. All units and molecular formulae should be in upright font. For example, $CO_2$ and $CH_4$ on p. 11, l. 8-9.

- Please number all subsections (with a boldface title). Now it is confusing that some are numbered and others are not.

- Please remove the historical references to Beer on the extinction law and to Junge on the power law size distribution. This is now all standard textbook material.

- Eq. 7 - 8: remove the unnecessary brackets around the gases in the subscripts, and remove the A, since A stands for aerosol.

- Please check the plurals: This values, etc.

---

## Referee Comment (RC4) · Anonymous Referee #4 · 15 Nov 2016

Ocean-based radiation measurements of any type are relatively rare, so it's always good to see a study trying to do just that. There is one issue with shipborne measurements (shadowband type) that I've never heard discussed, and that is how confident one can be in identifying the precise sun-obscured moment from each shadowband pass? For clear sky conditions the exorcise is straight-forward. For very overcast conditions one might follow the authors' example and only provide global data. What about everything in between? It's not hard for me to imagine sky conditions that obfuscate the actual sun-obscured moment, and lead the algorithm to an incorrect determination. I'll admit to never having worked with shipborne measurements, but it seems to me it

would be important to develop an algorithm that compares each measurement with the preceding and subsequent data point(s) as a way to gain confidence in the exact timing of the moment when the sun is completely obscured. Such a test could be developed using clear sky data with the goal being to produce a confidence level for the timing of when the sun is blocked during each measurement set (or sweep). If this issue has been dealt with adequately in a prior paper, then please provide some text along with a reference.

A shadowband instrument is presented, along with land-based data, yet there are no Langley calibrations presented. A long enough time series of Langley cals might show a temperature dependency that could be used to further improve data. (I do understand the instrument is temperature stabilized).

Uncertainties are given to two decimal places throughout the paper. One decimal at most for this work.

In the introduction the instrument is described as having "a constantly moving shadow band" (P3 L1). From the instrument picture (Figure 1), which, BTW, is an exceedingly poor picture, it's obvious the shadowband (one word) cannot move continuously. Later in the manuscript the shadowband motion is described as "sweeping" which sounds more accurate. Are measurements made in each direction, or does the shadowband always return home after a measurement set? How often are measurements taken? Is the frequency fixed or user configurable?

So in an effort to learn more about how this instrument operates I looked to the Seckmeyer et al., 2010 reference (P3 L16) as the manuscript strongly implies it to be a description of the instrument. It's not. Is there a peer-reviewed reference that describes this instrument in detail? Preferably with the BioSHADE accessory.

P3 L2 Should be channels, and "...includes all AEORNET and MFRSR channels." I would say rather it includes five channels that are very close to standard MFRSR channels and one that matches exactly (940). I cannot say if similar wording changes

should made in respect to CIMELs.

P3 L22. At this location in the manuscript are the authors asserting the 18 channels are measured simultaneously?

P4 L8. black anodized not "anodized black."

P4 L9. "when the band is moving" to "during a measurement sequence." This goes back to the earlier statement that the band "is constantly moving." Also "rotates". To me rotation implies 360deg. I like the use of "sweep" better as is done later in the manuscript. It better describes the movement of the shadowband.

P4 L10. Band can't be "stowed" if it's constantly moving.

If there isn't an authoritative article on the GUVis-3511 then this section needs significant improvement. Also, there is no mention of why the band width is 2.5 cm and the diameter 26.7 cm. I'm hopeful there is a better picture of the instrument, preferable taken from slightly above the sensor.

P4 L24. extent not extend

P4 L31. "To improve stability..." sentence is poorly worded.

P5 L4. Is it possible to load one's own calibrations?

P6 L3. airplane is one word.

P6 L10. These factors...

P7 L2-4. Are the internal measurements of "pitch and roll" applied internally or during post-processing? Are these data part of the datastream? Consider using x-axis and y-axis for land-based situations as pitch and roll are ship/aircraft terms.

P7 L19. Figure 4 demonstrates... I read L19 - L21 many times. I now think I understand what is being conveyed, but the passage is confusing.

P7 L22. From these calculations...

P7 L27. In this section detecting the minimum when skies are clear, or at least the sun is not obscured, and what to do when direct irradiance is very small are both discussed. There are many situations in between these extremes that are not addressed at all. I see this as a major deficiency. If the paper were only on AOD and direct beam that's one thing, but the opening sentence of the abstract promises us "shipborne (one word) measurements of the direct, diffuse and global spectral irradiance components..."

P10 L25. uncertainties are...

P11 L1. From these values...

P11 L22. ...reaches up to 0.5. The ending of that sentence leaves me hanging.

P10 L12. ...therefore been excluded

P10 L22. ..with an uncertainty

P14 L15. Why should I believe the Microtops II is an instrument worthy of making a claim the GUVis compares well with? The first referenced article in this section (Macke, 2009) only briefly mentions the Microtops, focusing mostly on its operation. The second referenced article (Smirnov et al., 2002) doesn't reference the Microtops II in the text at all. There is nothing here to give the reader confidence the Microtops II is anything more than an instrument that provides the operator a general idea of AOD. And actually, I don't understand the Smirnov reference in the context of the text at all.

P16 L24. ...the fact that the...

Fig 1. How about... GUVis mounted on research vessel Polarstern during cruise PS83. A total sky imager is to the left.

Fig 2. ...are in yellow... ... are in green...

Fig 3. How about... Figure shows factors for motion correction measurements of 305 nm and 510 nm GUVis channels. Existing caption is unnecessarily wordy. Why say "Additionally" when the opening sentence states this figure shows correction factors?

"By adapting...into account" is superfluous.

Fig 6. Relationship of precipitable water obtained from CIMEL sun photometer and GUVis shadowband radiometer during Melpitz-Column experiment.

Figs 7-13. Often more text than necessary.

Fig 11. This figure doesn't present the data clearly. Consider a different approach. Maybe plotting the differences?

Fig 12. I'm not sure what is being presented?
* * *

---

## Referee Comment (RC5) · Anonymous Referee #5 · 19 Nov 2016

The manuscript "Ship borne rotating shadow band radiometer observations for the determination of multi spectral irradiance components and direct sun products for aerosol by Witthuhn et al. requires significant revision before it can be considered for publication. The manuscript is interesting and relevant, it just needs to be rewritten.

The manuscript has been carelessly prepared and that severely detracts from what should be the message of the paper. Rather than provide a detailed review, at this point, I would prefer to point out three specific examples that form the basis of my opinion that the paper needs significant revision before it should even be considered

for publication.

Starting with what should be a straightforward description of the instrument, the GU-Vis instrument is described in the abstract as "The 19 channel rotating shadow band radiometer. . ." while the instrument description on p.3 (line 18, section 2 Instrumentation) states that "GUVis radiometer is a multi channel filter instrument. . .with 18 spectral channels"

In the Introduction to the paper on p.2 line 20, the authors state "In addition, it provides direct information about radiative fluxes. . ." this statement is not true. The instrument does not measure fluxes.

Similarly, the statement (line 25 of the Introduction p. 2) "The simultaneous measurements with the shadow band radiometer of aerosol optical properties and radiative fluxes avoids inconsistencies in calibration which are unavoidable if multiple detectors are used. Aerosol size distributions can be obtained from the spectral dependence of AOD. . ." is also untrue. First, I have already objected to the term "flux" to describe the measurement. Second, while I agree that calibration would be more of an issue with multiple detectors the instrument contains multiple filters that are more of a problem for spectral calibration than multiple detectors would be. Third, the instrument uses multiple detectors: namely silicon photodiodes are used for wavelengths up to 1020 nm while indium gallium arsenide detectors are used at longer wavelengths.

While revising the manuscript, the authors should consider expanding their discussion of the cosine correction. On p. 7 (lines 6-10) they state that they are using the measurements provided by the manufacturer. They should specifically examine their data for errors in this cosine characterization which should be filter dependent and therefore introduce a spectrally-dependent source of error/uncertainty which would show up in an examination of daily time series of retrieved aerosol properties and add some discussion to Section 4.4 (Discussion of the Uncertainty). Doing this would allow the authors to separate the spectral uncertainties due to errors (uncertainties) in the characterization of the filter response function and errors (uncertainties) in the characterization of the cosine response of the filters.

---

## Author Comment (AC1) · 31 Jan 2017

The final response was uploaded in the form of a supplement. Included are the response letter to all reviewer "authors_response.pdf", the revised paper "paper_revised.pdf", and the latex diff to the previous version of the paper "latex_diff.pdf".

Please also note the supplement to this comment:
http://www.atmos-meas-tech-discuss.net/amt-2016-297/amt-2016-297-AC1-supplement.zip